# An Assessment of Hydroacoustic and Electric Fishing Data to Evaluate Long Term Spatial and Temporal Fish Population Change in the River Thames, UK

**Jim Lyons [1],\*, Jon Hateley [2], Graeme Peirson [3], Frances Eley [4], Stuart Manwaring [5] and Karen Twine [1]**

[1]  Environment Agency, Trentside, Nottingham NG2 5FA, UK; karen.twine@environment-agency.gov.uk
[2]  Environment Agency, Richard Fairclough House, Warrington WA4 1HT, UK; jon.hateley@environment-agency.gov.uk
[3]  Environment Agency, Mance House, Kidderminster DY11 7SL, UK; graeme.peirson@environment-agency.gov.uk
[4]  Environment Agency, Rivers House, Colchester CO5 9SE, UK; fran.eley@environment-agency.gov.uk
[5]  Environment Agency, Red Kite House, Wallingford OX10 8BD, UK; stuart.manwaring@environment-agency.gov.uk
\*  Correspondence: jim.lyons@environment-agency.gov.uk; Tel.: +44-(0)208-4746-712

**Abstract:** This paper reports the results from mobile hydroacoustic surveys carried out between 1994 and 2018, to assess the fish stocks in four impounded reaches, covering 19.8 km of the River Thames, England. The data are complemented with electric fishing boom boat results, collected at the same study reaches and time periods. Hydroacoustic surveys used inter-calibrated dual and split-beam scientific echosounders, with the transducers beaming horizontally across the river to provide fish abundance and distribution estimates. Electric fishing surveys provided catch per unit effort estimates and information on size structure and species composition. Catch data from the margins of the study reaches were dominated by roach (*Rutilus rutilus*), with bleak (*Alburnus alburnus*) dominant in mid-river. Hydroacoustic data demonstrated patchy spatial distribution, often associated with natural and anthropogenic habitat features. Cyclical peaks and troughs in both hydroacoustic and electric fishing abundance were found. There were periods of correspondence with electric fishing abundance estimates, but also periods of significant divergence. The concept of 'Shifting Baseline Syndrome' is discussed with reference to these data, illustrating the importance of viewing long term quantitative information when using meaningful reference points. The potential impact of river temperature and flow on the fish population data are discussed.

**Keywords:** River Thames; hydroacoustics; electric fishing; long term data; fish stock assessment

## 1. Introduction

This study represents the longest single continuous and standardised application of horizontal hydroacoustic methodology for monitoring a lowland river's fish populations. These hydroacoustic data are a comprehensive long-term baseline of fish abundance and distribution, based entirely on non-destructive sampling. This information is complemented with species composition derived data from a long-term time series of electric fishing surveys, collected with the same sampling frequency and collection dates. These longstanding data provides a unique opportunity to investigate temporal and spatial shifts in the fish population on a world-renowned river coarse fishery.

The River Thames lies in a predominately lowland catchment with a floodplain area of 896 km². Within the UK, freshwater angling is one of the most popular sports, with large lowland rivers an important resource [1,2]. The River Thames is a valuable socio-economic angling venue with annual local licence sales of approximately 250,000 and £5

million revenue [3]. Over 15 million people live within the Thames catchment resulting in significant environmental pressure on the river, and potable water demand in the upper Thames is primarily managed by Thames Water Utilities Ltd. (TWUL). Population growth in the catchment and the associated pressure on water supply have, over the years, caused TWUL to investigate several strategic resource options.

In 1976, Thames Water Authority proposed the construction of a new reservoir at Abingdon to meet an increased demand for drinking water. The reservoir will allow more water to be abstracted from the river at times when water is available. It can then be stored prior to treatment for potable water supply. There is also scope to permit water to be re-introduced into the river at times of low flow. In response, government environmental regulators committed to environmental impact baseline assessments. This established the Thames Water Abingdon Reservoir Programme (TWARP), with hydroacoustics identified as a key survey method for fish population assessment. Since 1994, annual hydroacoustic surveys have been conducted on the river from Abingdon to Benson. Another proposal is the transfer and discharge of raw and treated water from other catchments into the River Thames. One of the proposed water transfer discharge points is within the area of study, located near to Culham. The transfer of water will augment flows in the River Thames in times of elevated pressure on water resources, such as prolonged dry weather events. To provide a baseline of fisheries data, a series of hydroacoustic and electric fishing surveys were commissioned by the local National Rivers Authority (NRA), latterly Environment Agency (EA), fisheries team.

Large (>20 m wide; >2 m deep) rivers are difficult to survey with conventional methods, such as electric fishing and netting. These methods can be adapted, for example boom boat electric fishing, to effectively sample marginal habitat. However, a quantitative assessment of the pelagic mid-water channel remains a challenge. Alternative capture methods commonly used worldwide, such as gill netting, are not an option due to an Environment Agency presumption against the use of gillnets by staff, based on the destructive nature of the method. Hydroacoustics offers an accepted method [4,5] to quantitatively survey mid channel open water habitat, found extensively in the navigable channelised middle reaches of the river. Over the last three decades hydroacoustics has increasingly been implemented worldwide to survey both river [6–9] and lake [10–16] freshwater fish populations, due to its ability to efficiently and non-invasively sample large volumes of water and habitats not suited to traditional fish capture methods [17,18].

Mobile hydroacoustic sampling in large UK rivers using a horizontally orientated transducer has been used to quantify fish populations since the 1990s [19–21]. In the UK, a national fisheries mobile hydroacoustic survey programme on large rivers was implemented in 2002. Prior to this, a decade of research and development was conducted with initial studies focused on the River Thames [19,22].

While hydroacoustics generate mass data on fish density, there is an absence or limit on species identification. In lakes with restricted species diversity, some sampling strategies have been developed [23], with target strengths related to species-specific behaviour and spatial positioning [24] to increase knowledge on community composition. This approach for horizontal sounding in large lowland rivers with a multi-species community is not viable and complementary methods are required to acquire biometric information on the fish population [4]. Electric fishing is a commonly used tool in fisheries [25–27] and the preferred method for river fish surveys in the UK. To obtain a full picture of fish stock status the current study recognised the need to use complementary methods for validating the data and address method-specific information gaps. It is essential that data acquired from capture and non-capture methods within the same monitoring programme are integrated effectively [28,29]. Previous hydroacoustic studies have addressed data validation using electric fishing and other capture methods such as angling creel census [30]. On occasion, the removal of water from a survey location has provided an opportunity to check the accuracy of hydroacoustic abundance and biomass estimates [31].

Potamodromous freshwater fish need to disperse or migrate throughout the year to gain access to reproduction, feeding and refuge habitats to complete their life cycle [32]. The Thames reflects many rivers around the world that are affected by human activities and habitat fragmentation, structural homogeneity, channel straightening and impounding [33,34]. Distribution of fish populations will respond to annual environmental variation, even in such relatively impoverished habitats. Only by capturing data over an extended time frame will an understanding of spatial change and location residency be understood.

This paper reports the results from data collected from 1994 to 2018. Hydroacoustic data are supplemented with boom boat electric fishing surveys carried out during the same time period to provide information on species composition. Abiotic data on temperature and flow are used to provide environmental context. The data collected in this study are regarded to be of sufficient spatial and temporal extent to provide valuable broader information of response of fish populations in a lowland UK river, to flood events and other climate change impacts.

## 2. Materials and Methods

### 2.1. Study Area

This study was conducted on a 19.8 km section of the non-tidal River Thames (Figure 1) between Abingdon Lock (NGR: SU 50633 97109) and Benson Lock (NGR: SU 56866 93629); the reaches selected for this study include that most influenced by water company proposals, namely, the Abingdon to Culham reach.

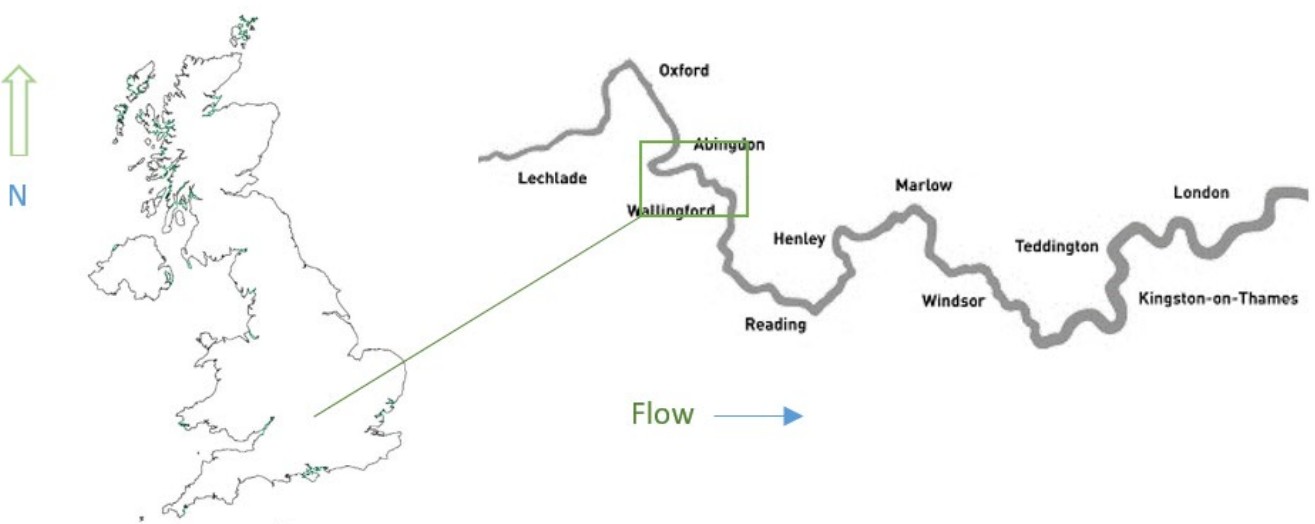

**Figure 1.** Location of the study area on the River Thames, England, UK.

The tidal limit at Teddington is 117 km downstream from Benson Lock. The area was divided into four reaches from Abingdon to Benson. Each reach was impounded by a navigation lock and weir at both its upstream and downstream limit. In downstream sequence, the reaches used in this study are: Reach 1: Abingdon—Culham (4.15 km) is the most upstream reach. Reach 2: Culham—Clifton (4.52 km); Reach 3: Clifton—Days (4.80 km) and Reach 4: Days—Benson (6.37 km).

The main river channel in this study is maintained for navigation purposes and has few natural hydro geomorphological features. Between Reading and Oxford, the EA, as part of its customer charter, maintains a minimum of 1.2 m depth over a navigation fairway. This is usually considered to be the middle third of the river or the approach to any of the lock sites, from an upstream of downstream direction. Bathymetry range for this section of the river is 2.11–7.34 m with an average depth of 3.37 ± 0.69 standard

deviation. River depth surveys were conducted by EA Geomatics using a RESON (Teledyne RESON A/S; Fabriksvangen 13, Slangerup, Denmark) Seabat 7101 Multibeam sonar. Vessel position and attitude was captured with an Applanix (Applanix Corporation; Richmond Hill, ON, Canada) POSMV-320 (S/N 3878) system controlled using a network of local Ordnance Survey OSNET stations. Using Ordnance Survey map readings at 500 m intervals, the average width of the river, with 95% confidence limits, was calculated at 44 m ± 2.0 m. Channelised for flood mitigation and navigation purposes, flow is predominantly glide with some marginal slack water. This confined river channel is largely separated from its floodplain except during 'out of bank' flood events. Four weirs (Abingdon, Culham, Clifton and Days) are present within the study area where turbulent air-entrained flow and deep pools provide a significant attraction to fish, in particular, rheophilic cyprinids [35] and predators [36]. Low signal: noise conditions immediately downstream of these structures prevents the use of hydroacoustics. Whilst electric fishing was conducted in or near weir-pools, collected data have been excluded to avoid spatial analysis bias between the two methods. Across all habitats the fish population is dominated by roach (*Rutilis rutilis*), bleak (*Alburnus alburnus*), perch (*Perca fluviatilis*) and chub (*Squalius cephalus*) with bream (*Abramis brama*), dace (*Leuciscus leuciscus*), barbel (*Barbus barbus*), gudgeon (*Gobio gobio*) and pike (*Esox Lucius*), also important.

### 2.2. Acoustic Sampling

The study period covers data collection from 1994 to 2018. Surveys were boat-based and mobile with a GPS derived ground speed of 5–6 km h$^{-1}$. The sound beam was oriented perpendicular to the boat's longitudinal axis. Surveys conducted from 1994 to 2002 used a BioSonics (BioSonics, Inc.; 2356 W Commodore Way, Unit 110, Seattle, WA, USA) model 102 dual-beam sonar operating at 420 kHz at 10 pings s$^{-1}$, with a pulse duration of 0.4 ms and 40 Log $R$ time-varied gain (TVG). A circular transducer with a 6° narrow and 15° wide beam was mounted on a Videmech (Videmech Ltd.; Yateley, Surrey, UK) pan and tilt head rigidly fixed 1 m in front of the boat and 0.8 m below water surface. From 2002 all data were collected using an HTI (Hydroacoustics Technology Inc.; 711 NE Northlake Way, Seattle, WA, USA) model 241 split beam system echosounder. This was operated at 200 kHz and 10 pings s$^{-1}$, with a pulse duration of 0.2 ms and 40 Log $R$ TVG. An elliptical split-beam transducer deployed with operating angles of 4° along its vertical axis and 10° in the horizontal axis was mounted in the same manner as the earlier BioSonics surveys. Inter-calibration of the two systems was conducted using field sample data from 2001 and 2002 when both systems were simultaneously deployed with the transducers on a common fixed mounting.

Biosonics echo signals were recorded onto a laptop and separately on DAT tape. Signals were monitored on the ESP oscilloscope screen in the PC and on a Phillips (Philips Electronics UK Limited; Farnborough, UK) PM97 'Scopemeter' oscilloscope. HTI echo signal capture was as RAW and BOT data files directly onto a laptop using the manufacturer's Digital Echo Processing software.

Before each survey period, the Biosonics equipment was calibrated following a standardised method [20]. This involved the use of a 21.4 mm diameter tungsten carbide sphere standard target of known acoustic target strength. For the HTI echosounder, standard target tests were conducted prior to each survey [37,38] using a 36.0 mm tungsten carbide sphere suspended >5 m from the transducer by monofilament line and fine mesh netting. Data from >250 echoes from each quadrant and >250 echoes on the acoustic axis were recorded and mean compensated TS calculated. If the mean was within ±3 dB of the theoretical TS of the calibration sphere (−39.5 dB), the equipment and associated calibration files were considered to be satisfactory and the survey proceeded.

River morphology and transducer attachment to a Videmech remote control rotator provided a maximum sample range of approximately 30 m; however, mean sampled ranges were generally much shorter. All surveys were confined to the hours of darkness

(+1 h sunset; −1 h sunrise), a time when fish were active in the water column and so detectable by horizontal sonar. During this time, boat traffic interference and bankside disturbance from anglers is at its lowest level. When required, a small spotlight was directed towards the nearest bank in the opposite direction of the sonar beam to aid safe navigation. Light attraction dispersal effects were assumed to be negligible due to the relatively low water transparency prevalent in rivers of this type.

Standard 10 min data files generated along each reach from 1994–2002 were analysed using the BioSonics Target Strength Post-Processing Program (ESPTS), containing the TS in decibels (dB) of the back scattering cross-section ± standard deviation for accepted fish targets (as single echo detections). For each surveyed reach, the run number and number of pings; the mean TS and standard deviation; the volume sampled and number of accepted fish targets were recorded. From these data, mean TS; standard deviation; total volume sampled; total number of fish detected; and fish density (fish 1000 m$^{-3}$) were calculated for upstream runs and downstream runs.

For HTI data collected from 2002–2018, standard 10 min data files were appended and processed using HTI Echoscape 3.00.10 software to generate single fish echo detections (dB) and sample volume (m$^3$). These files were subsequently analysed using Mobile Utility Analysis, a Microsoft Access program developed by the EA to apportion data into 100 m river length bins. Ping sample volume, total sample volume, number of pings, number of accepted echoes and fish density (fish 1000 m$^{-3}$) were calculated for each 100 m of river length within the study reach.

A −50 dB minimum acoustic threshold was applied to all data. A best fit linear regression (y (HTI) = 1.6396x (BioSonics) + 3.0592, $R^2$ = 0.3864) obtained for Reach 3 and Reach 4 in 2001 was applied to standardise the abundance estimates from the two systems [39]. All BioSonics abundance results are thereafter reported as HTI 'equivalents'. The relationship between acoustic size (dB) and real size (fork length in mm) was established using species (cyprinid and perch) and frequency appropriate (200 kHz) regressions [40]:

$$Y = aX + c \tag{1}$$

where $Y$ is TS (dB), $X$ is $\log_{10}$ length (mm) and $a$, $c$ are regression constants. For fish insonified in side aspect: a = 29.1966; c = −98.329. For fish insonified in mean aspect: a = 22.5811; c = −93.617. The calculated minimum 'visible' fish length were 40.7 mm and 85.4 mm for side-aspect and mean-aspect orientation, respectively.

### 2.3. Electric Fishing

Electric fishing surveys deployed a boom-boat arrangement (Figure 2) with pulsed direct current and output settings at 230 V, 50 Hz and 10–11 A. Power to a model FC3000GPBS control box (Electracatch International; Wolverhampton, UK) was supplied by a 7 kW Honda EU70is petrol generator. Output (230 V, max current 20 A) was then fed out to a pair of anode arrays on booms which sit at river surface level. The two cathode cables were fixed, one to each side of the boat, to stainless steel plates situated beneath the netting punts. Surveys involved timed runs carried out in a downstream direction. Surveys were conducted in the river margins and at mid-river locations and started around dusk each night. The results are presented as catch per unit effort (number per minute) estimates. All captured fish were identified to species level and fork length measured to the nearest mm. Within each reach, and for each species, scale samples from a minimum of three individuals per 10 mm size band were taken for age and growth analysis.

The mid-river boom-boating terminated in 2006, as these yielded fewer fish of fewer species and it was considered a relatively inefficient capture method. From 2006 onwards, margin electric fishing changed from a series of five-minute runs in each reach, to a recording of total fishing time. This is because increasing riparian tree cover often precluded continuous fishing for a five-minute period; the fishing interrupted as the boat was steered into mid-river to avoid protruding trees.

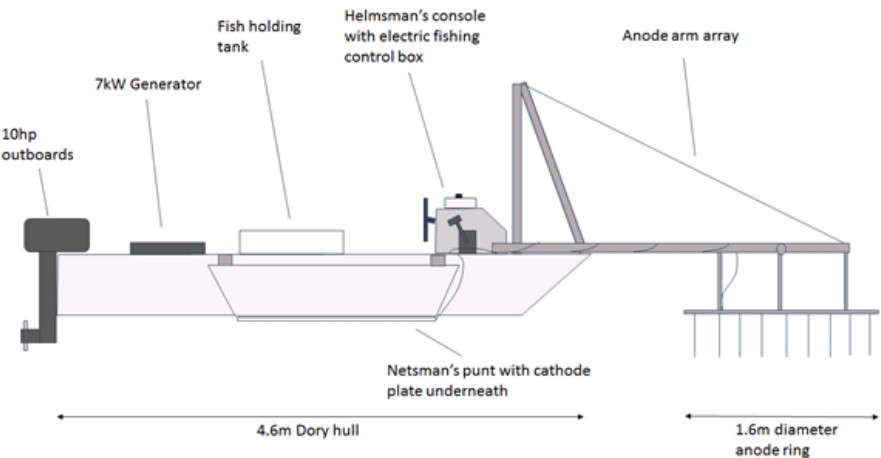

**Figure 2.** The design and arrangement of the boom boat used on the River Thames during the study period.

*2.4. Abiotic Data*

Habitat features, both natural and anthropogenic, with an obvious channel interrupting character were recorded using aerial survey and Ordnance Survey© map information. Channel habitat interrupting features (CHIF) are categorised as bridges, river tributary confluences, lotic off-river features (marinas and ORSUs), navigation lock channel confluences, weirs, sluices and islands.

Daily river temperature (°C) and flow ($m^3 s^{-1}$) for the study period were acquired from long term environmental monitoring assets on the river. Temperature data were provided by the UK Centre for Ecology and Hydrology (CEH) from their Wallingford station (NGR: SU 60900 90200). Temperature was measured using an ATP multi-thermo digital thermometer (ATP Instrumentation Ltd; Ashby-de-la-Zouch, UK). Information on flow and river levels was from the EA hydrometric gauging station at Sutton Courtenay (NGR: SU 51710 94619). To establish river water clarity levels, present during hydroacoustic and electric fishing surveys, routine data on turbidity were acquired. Water turbidity data are routinely measured by the EA in Nephelometric Turbidity Units (NTU) at Cleeve Lock, the site approximately 10 km downstream from Benson Lock. In 2014 and 2015, between June and August, average NTU was 6.4 and 10.3, respectively. These values approximate to Secchi Disk depth readings of 50–70 cm [41].

*2.5. Data Analysis*

Temporal and spatial variation in fish abundance were assessed for both hydroacoustics and boom-boat electric fishing data. Hydroacoustic fish density (fish 1000 $m^{-3}$) was calculated at both combined section and individual reach levels. Electric fishing abundance data are presented using catch per unit effort (number per minute) calculated from timed runs.

Combined average annual density for the four reaches was calculated and a best fit 5th order polynomial regression applied:

$$y = ax^5 - bx^4 + cx^3 - dx^2 + ex - f \tag{2}$$

where y is fish density and x, years. Comparison of statistical variation between reaches for annual combined upstream and downstream transect fish density (fish 1000 $m^{-3}$) was calculated using ANOVA analysis.

Regression analysis on the average estimated acoustic density (single targets >−50 dB) for each reach was also conducted and a best-fit smoother line applied using a LOESS two-step quadratic analysis.

For electric fishing, data were combined across all reaches for all years. Fifth order polynomial best-fit regression lines were applied to both margin and mid-river locations

to indicate CPUE trend across the survey periods. For roach and bleak, the two most abundant species, best fit cubic regression analysis was conducted:

$$y = ax^3 + bx^2 - cx + d \tag{3}$$

where y is CPUE and x, years. These species are typically located in mid-water and are therefore more acoustically visible than benthic dwelling fish.

All acoustic density estimate values used in the study are geo-referenced in the form of latitude and longitudinal coordinates. This information provides x, y and z coordinates for mapping analysis. ArcGIS 10.4.1 (Environmental Systems Research Institute; Redlands, CA, USA) was used to model the location and value of all fish density data used in this study. Spatial interpolation was applied using an inverse distance weighting (IDW) three-dimensional surface raster contour model to create a continuous (or prediction) raster grid using density, latitude and longitude values. The modelling assumes that spatially distributed objects are spatially correlated with points close together having similar characteristics when compared to distant neighbours [42]. Predicted values were then assigned to locations within the raster dataset based on the measured value (fish density) of each data point and its linear distance to a defined number of nearest neighbours. Analysis was based on an exponential reduction in influence with distance, this applied to the nearest ten neighbourhood points. Contour lines connecting locations of equal value in the resultant surface raster dataset were then applied with an isoline spacing of 100 fish 1000 m$^{-3}$.

Habitat preference was assessed by plotting GIS contour clusters where fish density exceeded 500 fish 1000 m$^{-3}$. Previous analysis on hydroacoustic data for the River Thames [43] using the method of [44] established an elementary distance sampling unit (EDSU) of 100 m. This distance is large enough to avoid an auto-correlative interpretation of the data whilst small enough to capture the main spatial structure of the fish population. High density clusters (HDC) within 100 m were treated as a single 'preferred' location. HDC-HDC centroid distance was calculated by using a geodesic calculation for the distance between two points on the surface of a spheroid [45]. Where more than one HDC was present within 100 m the IDW derived cluster with the greatest fish density was used to establish a single point location.

ANOVA and T-test analysis was used to establish the presence of any statistically significant difference for temperature and flow between years with peak hydroacoustic fish density estimates and long-term average (LTA) data.

## 3. Results

### 3.1. Temporal Variation in Fish Density along the Study Section

3.1.1. Hydroacoustics: Cyclical Variation, +/− LTA, Density Comparison with other Rivers

A total of 5310 acoustic data files were analysed for the study period (1994–2018). These files represent the raw survey field data collections prior to post processing and analysis.

Average (± 95% confidence limits) echo-counted hydroacoustic density for the study period was calculated for each reach (Reach 1 = 93.07 ± 8.81; Reach 2 = 87.16 ± 7.57; Reach 3 = 83.62 ± 5.96; Reach 4 = 50.49 ± 2.94 fish 1000 m$^{-3}$). These values compare with the long-term average for mobile horizontal hydroacoustic surveys on large lowland UK Rivers of 53.63 ± 21.53 fish 1000 m$^{-3}$ [46].

Trend analysis of the fish density data (Figure 3) was conducted by applying a 5th order polynomial best-fit trend line to fish density average (±95% confidence limits). The coefficient of variation for this trend line was calculated as 0.3642, indicating that the data describe approximately 36% of the variation noted. The trend in variation describes a cyclical pattern of change during the study period with a peak to trough periodicity of approximately 6–7 years. Peak abundance was from 2008 to 2010 with relatively high

abundance also occurring in 1995, 2003 and 2015. In contrast, lowest annual average density was recorded in 2001, 2002, 2013 and 2017.

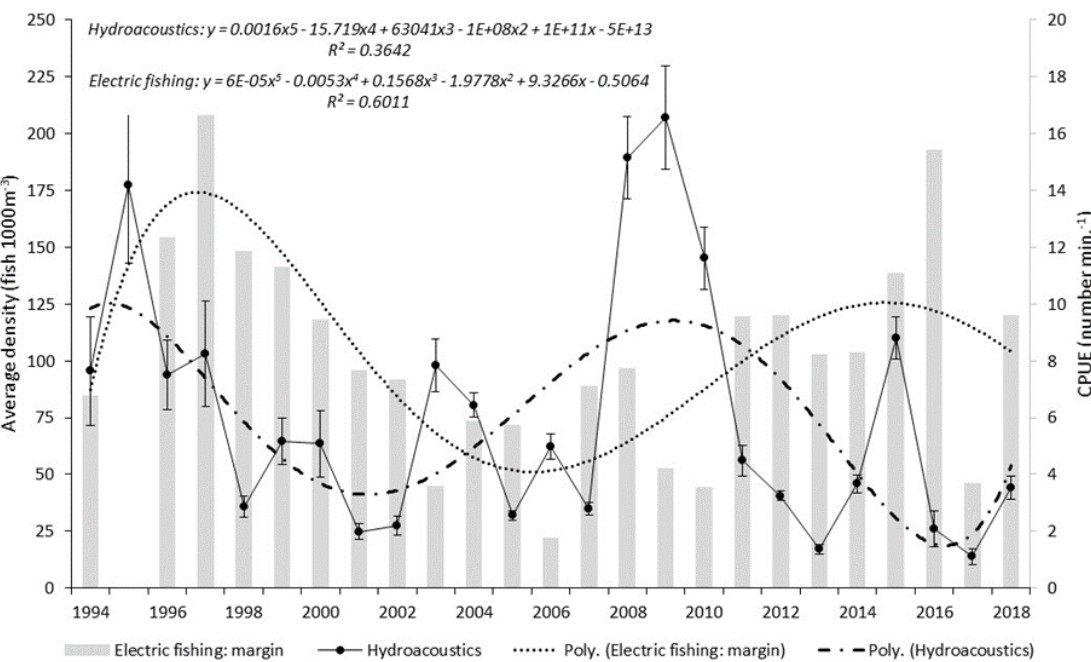

**Figure 3.** Variation in average hydroacoustic fish density (±95% confidence limits) and marginal electric fishing catch per unit effort (number min.$^{-1}$) sampling for bleak and roach during the study period (1994–2018) for combined reach data with best fit 5th order polynomial trend lines and calculated coefficient of determination applied.

### 3.1.2. Electric Fishing: Species and Abundance Shift/Variation

Roach and bleak were the dominant species, numerically, in the study section for both margin and mid-river boom-boat electric fishing surveys. The average contribution across all years of each species to the proportion of the total captured fish (number min$^{-1}$) varied with capture location. To avoid temporal bias in species abundance, all data were standardised to the same time period using the maximal data available for both margin and mid-river surveys. For surveys conducted in the river margins, bleak contribute 17.17% and roach 60.87%, to the total catch. Bleak and roach abundance for surveys conducted in the mid-river were 60.67% and 29.4%, respectively. The only other species that contribute greater than 2% to the captured population were chub and perch. In the margin samples, chub and perch contributed 4.29% and 9.20%, respectively. Chub and perch contributed 2.34% and 2.63%, respectively, to samples collected from mid-river surveys. River margin electric fishing surveys revealed species average size (fork length, mm) based on measured individuals: bleak = 85.41 ± 2.89 (Range = 20–192; $n$ = 2531); roach = 116.52 ± 3.99 (Range = 20–337; $n$ = 8975); chub = 186.17 ± 18.86 (Range = 38–550; $n$ = 633); perch = 145.14 ± 7.04 (Range = 41–445; $n$ = 1356). For mid-river electric fishing surveys (1994–2005) species average size (fork length, mm) based on measured individuals was: bleak = 90.09 ± 2.46 (Range = 21–196; $n$ = 7254); roach = 132.86 ± 6.05 (Range = 27–347; $n$ = 3515); chub = 249.68 ± 22.09 (Range = 22–534; $n$ = 280); perch = 164.33 ± 12.66 (Range = 44–371; $n$ = 314).

Combined section variation in all-species abundance (CPUE) showed temporal cyclical change (Figure 4). Separate plots for surveys conducted in the river margins (1994–2018) and mid-river (1994–2005) illustrate the variation in fish capture between electric fishing locations. No electric fishing surveys were carried out in 1995. Results for river margin surveys show a bimodal distribution with peak abundance in 1997 and 2016. From 2004 to 2010, abundance is at a minimum. Data from the relatively time-limited mid-

river surveys show a unimodal distribution, in which abundance peaked in 2000 and 2001 with 2002–2005 having the lowest recorded values.

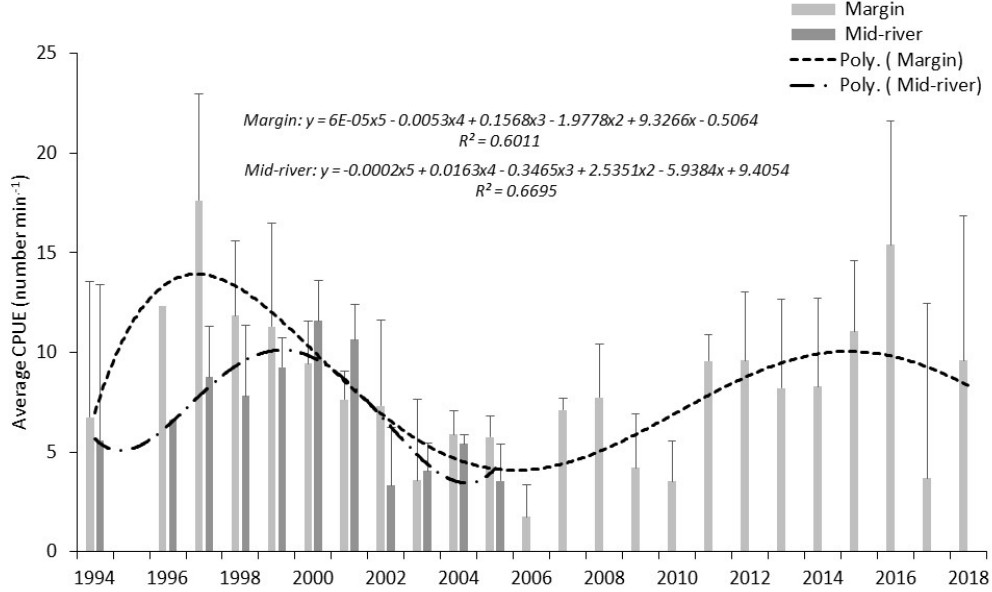

**Figure 4.** Combined all reach variation in average annual catch per unit effort (number min$^{-1}$; upper 95% confidence limit) for margin and mid-river boom-boat electric fishing surveys conducted from 1994 to 2018 with best fit 5th order polynomial trend lines and calculated coefficients of determination applied.

Electric fishing boom boat results for combined section and survey location (margin and mid-river) analysis for roach and bleak are similar to an all-species analysis reflecting the dominant abundance of bleak and roach throughout the survey area. Bleak show a gradual decline in abundance (CPUE) from 2004 to 2009. From 2010, relative abundance increases sharply, reaching peak levels for the last year of the study period in 2018. A similar pattern of abundance (CPUE) change is seen for roach although the decline from 1994 reaches a minimum in 2007. After 2007, a continual increase in abundance is seen and, as with bleak, reaches a maximum for the study period in 2018.

### 3.2. Comparison of Annual Average Abundance Estimates for Hydroacoustics (Fish 1000 m$^{-3}$) and Electric Fishing (Number Min$^{-1}$)

Results from the two fish survey methods applied in this study show a clear difference in the time series (1994–2018) for measures of fish abundance (Figure 3). Hydroacoustic annual average abundance shows a decline in both the early years (1994–2002) and later years (2011–2018) of the study period. During the defined later years period, a single year increase in acoustic abundance was noted in 2015. The intervening period records elevated abundance levels reaching a peak in 2008–2009. In contrast, electric fishing shows elevated abundance for both early (1996–1999) and later (2011–2016) years with a corresponding low abundance period (2003–2010).

### 3.3. Longitudinal Variation in Hydroacoustics Fish Density along the Study Section

An Anderson–Darling normality test of data by reach and its transformation for subsequent one-way ANOVA analysis of the annual fish density means was conducted For all reaches, the null hypothesis of normality is rejected for the raw data ($p < 0.05$) (Table 1). For the raw data, a log$_{10}$ (x + 1) transformation was applied to achieve normality. The resultant normalised reach data was then explored for a range of survey periods to test for homogeneity of means (Table 2). When applied to the entire survey data period (1994–2018), a one-way Anova test indicated no significant difference between reach mean

values. With F < F$_{crit}$, we can accept H$_0$: R1 (log x$^-$ +1) = R2 (log x$^-$ +1) = R3 (log x$^-$ +1) = R4 (log x$^-$ +1). Further analysis focused on the period of maximal acoustic fish density (Figure 2). No statistical significance between reaches at $p$ = 0.05 was found when the same analysis was confined to the survey period 2008–2010. However, when applied to data for 2009–2010, F > F$_{crit}$ and $p$ < 0.05 resulted in a rejection of the null hypothesis (H$_0$). confirming that for this data selection, a significant difference exists between reach mean density.

**Table 1.** Anderson–Darling normality test results for average fish density at all four study reaches.

| Test | Reach 1 | Reach 2 | Reach 3 | Reach 4 |
|---|---|---|---|---|
| A-D$^2$ | 1.94 | 1.62 | 1.16 | 1.37 |
| $p$ | <0.05 | <0.05 | <0.05 | <0.05 |
| A-D$^2$ log$_{10}$ (x + 1) | 0.14 | 0.38 | 0.20 | 0.25 |
| P log$_{10}$ (x + 1) | 0.97 | 0.39 | 0.87 | 0.72 |

**Table 2.** One-way ANOVA results comparing reaches from three survey periods for average fish density with an applied log$_{10}$ (x + 1) transformation.

| Survey Years (All Reaches) | df | F | F$_{crit}$ | $p$-Value |
|---|---|---|---|---|
| 1994–2018 | 99 | 0.642 | 2.699 | 0.589 |
| 2008–2010 | 11 | 3.232 | 4.066 | 0.820 |
| 2009–2010 | 7 | 7.179 | 6.591 | 0.044 |

*3.4. Comparison of Estimated Fish Densities Between Reaches*

Having established a period of clear spatial variation within the river at a reach scale, temporal analysis of data was conducted to establish if between year variation was present. Temporal deviation is greatest for the 2009–2010 period with Reach 4 showing a lower response in increased fish density when compared to Reaches 1–3 (Figure 5).

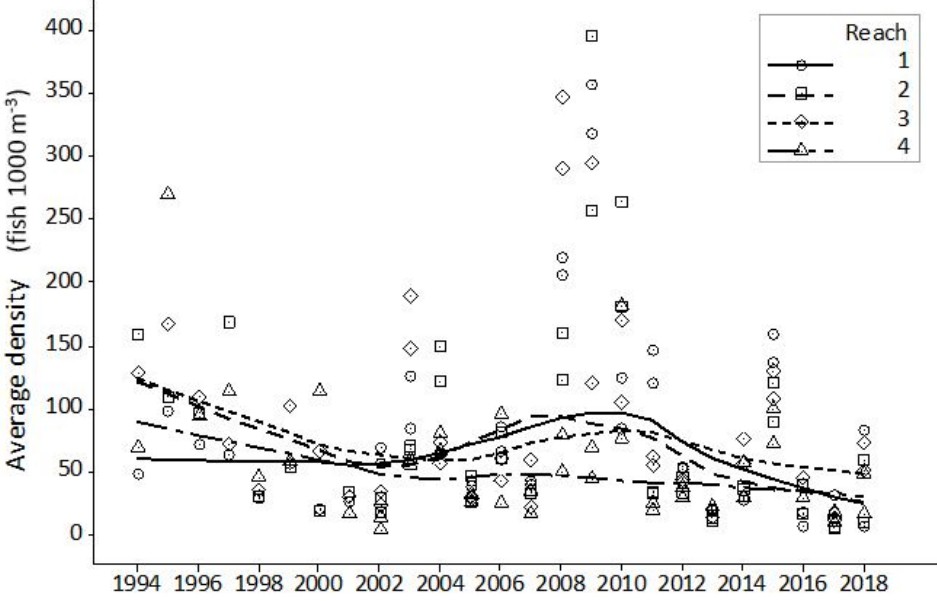

**Figure 5.** Annual average fish density with a LOESS fitted smoother line (0.5; 2 step; quadratic) for each reach.

*3.5. The Effect of Habitat Variation on Fish Density along the Study Reach*

Hydroacoustic long term average density for the study area (1994–2018) was 75.58 ± 22.65 fish 1000 m$^{-3}$.

Habitat preference was assessed by plotting GIS contour clusters where fish density exceeded 500 fish 1000 m$^{-3}$. A total of 40 HDCs were identified based on IDW 3D cluster analysis where cluster centroid >500 fish 1000 m$^{-3}$. Applying a minimum elementary distance sampling unit (EDSU) of 100 m to the data resulted in a reduced number of 30 HDCs present in the study reach after applying this minimum spacing distance (Figure 6).

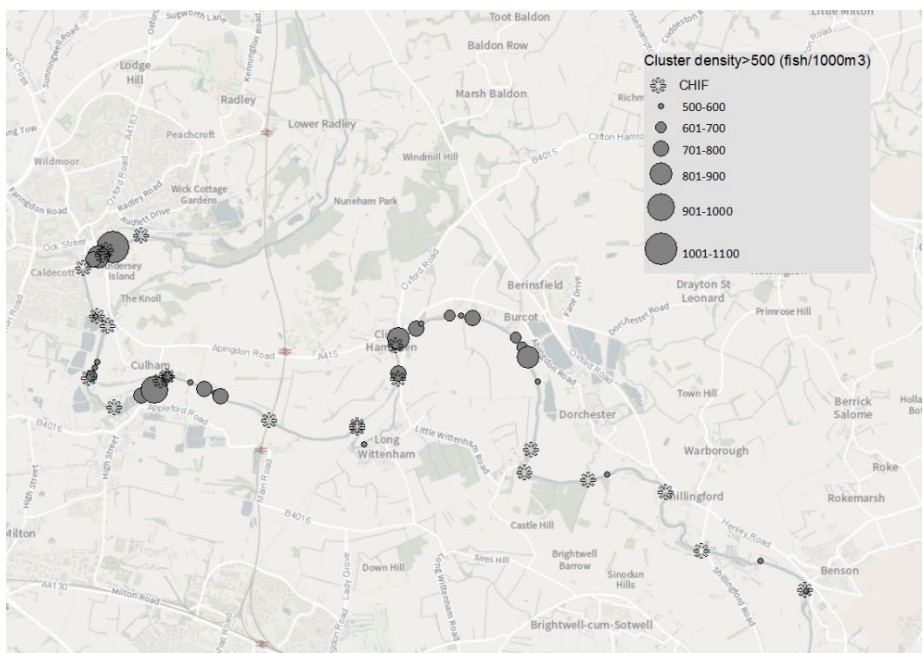

**Figure 6.** Centroid locations of contour cluster single target fish density (>500 fish 1000$^{-3}$) greater than 100 m centroid-centroid spacing derived from GIS modelled inverse distance weighted three-dimensional analysis and location of channel habitat interrupting features (CHIF) for Abingdon Lock to Benson Lock (1994–2018).

Location analysis of HDCs finds that the spread of clusters within the study area reveals an uneven distribution with Reach 3 (*n* = 13) recording the highest number of clusters and Reach 4 (*n* = 3) the fewest (Table 3).

**Table 3.** Location clusters from Abingdon Lock to Benson Lock with the highest mean (± 95% CL) acoustic fish density based on three-dimensional inverse distance weighted spatial analysis for the period 1994–2018.

| Reach | Cluster Density > 500 Fish 1000 m$^{-3}$ | Cluster Density Maximum (Fish 1000 m$^{-3}$) | Cluster Density (Mean ± 95%CL) |
|---|---|---|---|
| 1 | 7 | 1044.52 | 719.16 ± 139.64 |
| 2 | 7 | 974.04 | 680.39 ± 120.35 |
| 3 | 13 | 888.07 | 656.61 ± 51.82 |
| 4 | 3 | 579.18 | 543.95 ± 77.80 |

In Reach 1, HDCs were located predominantly upstream and downstream of Abingdon Bridge (NGR: SU 49957 96855). Other clusters were found immediately downstream of the US Culham Cut feature (NGR: SU 49767 94853). All clusters in Reach 2 were upstream of Appleford Railway Bridge (NGR: SU 52657 94166), primarily located upstream and downstream of DS Culham Cut (SU 50999 94889) confluence. The

remaining location with significant HDCs were in Reach 3 between Clifton Hampden Bridge (SU 54715 95381) and adjacent to Orchid Lakes, Burcot (NGR: SU 57113 95502).

A total of 21 significant channel habitat interrupting features (CHIF) were identified within the entire study reach: Type I: Bridges (*n* = 5); Type II: River tributary confluences (*n* = 2); Type III: Lotic off-river features (n = 2); Type IV: Navigation lock and backwater channel confluences (*n* = 7); Type V: Weirs and sluices (*n* = 4), islands (*n* = 1). A total of 12 HDCs (Reach 1 = 6; Reach 2 = 2; Reach 3 = 3; Reach 4 = 1) showed a close spatial association with a CHIF, being within 100 m of the identified habitat feature (Table 4). The remaining 18 HDCs were more than 100 m distance from any CHIF.

**Table 4.** Reach habitat analysis showing the number of high-density clusters (HDC) <100 m of a channel interrupting habitat feature (CHIF).

| Metric | Reach 1 | Reach 2 | Reach 3 | Reach 4 |
|---|---|---|---|---|
| HDC (Type I CHIF) | 1 | 0 | 2 | 0 |
| HDC (Type II CHIF) | 1 | 0 | 0 | 0 |
| HDC (Type III CHIF) | 1 | 0 | 0 | 0 |
| HDC (Type IV CHIF) | 3 | 2 | 1 | 1 |
| HDC (Type V CHIF) | 0 | 0 | 0 | 0 |
| Total | 6 | 2 | 3 | 1 |

Fourteen of the CHIFs had no HDC within 100 m (Table 5).

**Table 5.** CHIFs without an associated (<100 m) HDC.

| Metric | Reach 1 | Reach 2 | Reach 3 | Reach 4 |
|---|---|---|---|---|
| CHIF (Type I): absent HDC | 0 | 2 | 0 | 1 |
| CHIF (Type II): absent HDC | 1 | 0 | 0 | 1 |
| CHIF (Type III): absent HDC | 0 | 0 | 0 | 1 |
| CHIF (Type IV): absent HDC | 2 | 1 | 1 | 1 |
| CHIF (Type V): absent HDC | 1 | 0 | 1 | 1 |
| Total | 4 | 3 | 2 | 5 |

*3.6. The Impact of Temperature and Flow on Fish Abundance*

River flow, levels and temperature were examined for potential impact on recorded fish densities. High summer flows on the Thames are unusual; however, particularly high flow events occurred in 2007 and 2012 between June and August (Figure 7). Only the 2007 event was of sufficient river water level height at 4.44 m above stage datum (ASD) to cause 'out of bank' conditions. Following the 2007 peak, high mean hydroacoustic fish densities were recorded in 2008–2010. However, no such peak immediately followed the 2012 summer event.

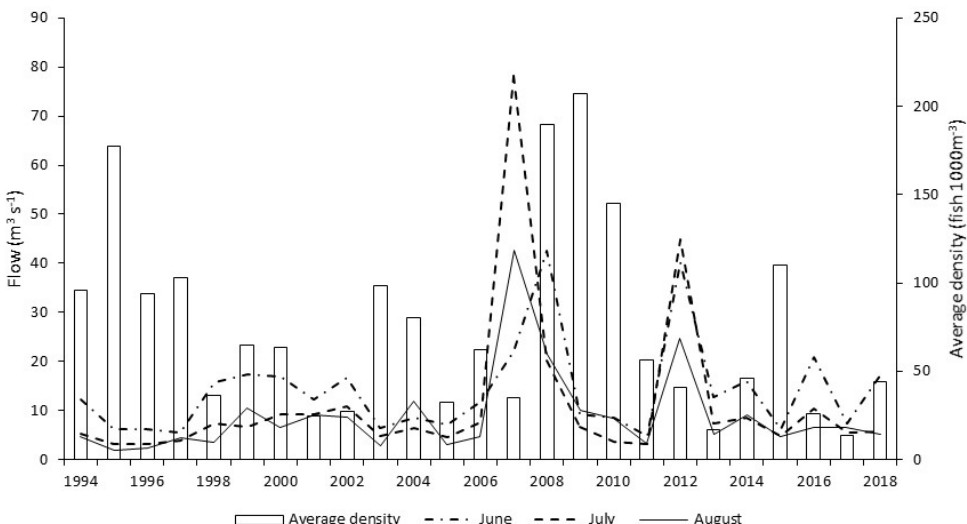

**Figure 7.** Graph showing the variation in annual hydroacoustic fish density (all reaches—fish 1000 m$^{-3}$) with mean daily flows (m$^3$ s$^{-1}$) for June, July and August from 1994 to 2018.

One-way ANOVA analysis comparing temperatures in 2007 and 2008 with 2012 and 2013 during the warmest months, when growth and recruitment are maximal (May–September), revealed no significant difference ($p > 0.05$; df =18) between periods. ANOVA analysis comparing the main overwintering periods (November–March) for 2007/2008 and 2012/2013 and the long term average (LTA) for the same months revealed a statistically significant difference ($p < 0.05$; df = 13) in water temperature (Figure 8) between the periods. The earlier period was found to be statistically similar to the LTA, with the later period significant colder than the LTA. Confirmation was provided by conducting a t-test comparing temperature for 2007–2008 with 2012–2013 only for this overwintering period which showed a statistically significant difference ($p < 0.05$; df = 5).

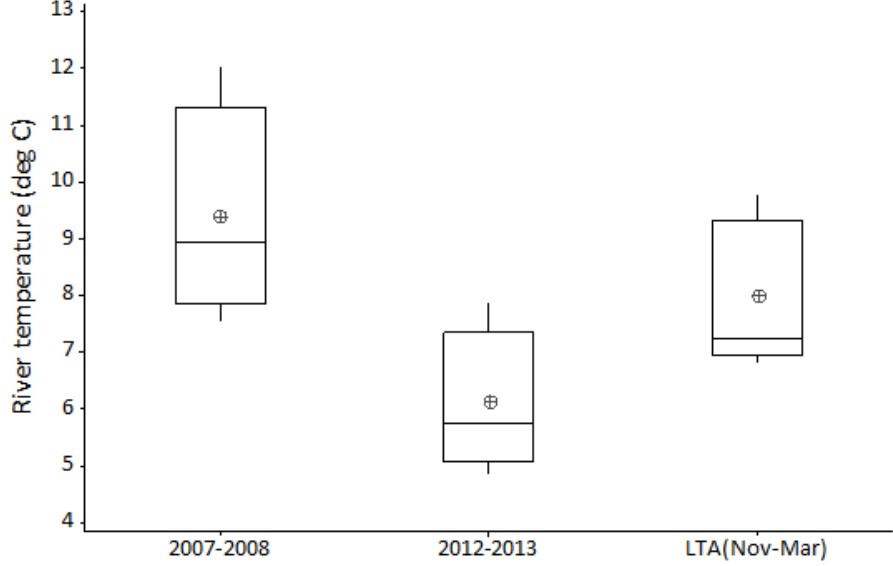

**Figure 8.** Variation in temperature (maximum, minimum, median, mean, 1st and 3rd quartiles) between two overwintering (November–March) time periods after high summer flows and the long-term average (LTA).

ANOVA analysis for flow to compare the same overwintering period revealed a statistically significant difference ($p < 0.05$; df =14) between monthly recorded LTA river flow and flows recorded in 2007–2008 and 2012–2013 (Figure 9). A *t*-test comparing monthly average flows for 2007–2008 with 2012–2013 for this overwintering period shows a statistically significant difference ($p < 0.05$; df = 7).

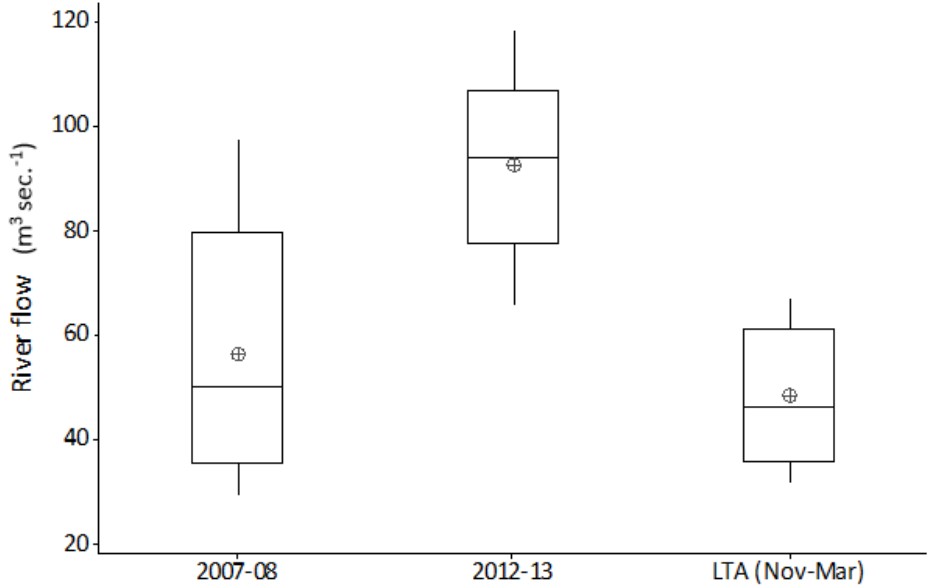

**Figure 9.** Variation in flow (maximum, minimum, median, mean, 1st and 3rd quartiles) between two overwintering (November–March) time periods and the long-term average (LTA).

One further difference between the two identified periods (2008–2010; 2012–2013) was floodplain inundation triggered by 'out of bank' river levels (river levels >3.51 m ASD at Sutton Courtenay, located in the middle of the study area, are regarded as 'out of bank' events). This occurred during the high summer flow event in 2007 and November and December 2012. To put this into context, during the entire study period, the river was out of bank on ten occasions. Of these, only once has this occurred outside of Nov–Mar and this was the July 2007 event when the highest recorded river level was 4.44 m ASD.

## 4. Discussion

### 4.1. Value of Long Term Regular Monitoring

Shifting baseline syndrome [47,48] occurs when conditions of the natural environment gradually degrade over time, yet local residents, natural resource users and policy makers falsely perceive less change because they do not know, or fail to recall accurately, how the natural environment was in the past. This may result from a lack of data on the natural environment, this study showing the value of consistent long term data collection and the potential errors in the extrapolation of limited temporal data.

Large rivers have a wide range of natural and anthropogenic environmental influences on the resident fish populations, which may result in spatial and temporal change in abundance. The density and distribution of lowland river fishes varies by season and time of day and is influenced by a range of abiotic, biotic and behavioral factors such as temperature, oxygen concentration, and vertical distribution of predators and prey [49]. For a monitoring programme to be effective, successful and sustainable over the longer term, it must not only be ecologically relevant and statisically credible, but also cost efficient [28]. The use of hydroacoustics in both this study, and elsewhere on large UK rivers, has shown the method to be both efficient and effective. The collection of quantitative data from up to 30 km of river in a single night, using a non-invasive

approach, is not possible with other fisheries survey methods. However, whilst hydroacoustic sampling is a powerful tool that delivers cost effective quantitative information on fish populations over many kilometres of river, it is not a universal panacea for large river monitoring, its main drawback being the absence of species information. To achieve this level of detail on large rivers and lakes, a multi-method approach is needed to both encompass the complex species–habitat interactions and spatially diverse fish populations. In this study, the use of boom boat electric fishing provided a suitable complementary method, providing valuable information on species composition, morphometric distribution and ageing analysis.

The TWARP hydroacoustics monitoring programme [50] that forms the basis of the current study conformed to both CEN and UK standards of mobile hydrocoustics monitoring [4,50]. With all surveys completed in July, this provided temporal consistency throughout the entire study period. In isolation, a single survey provides a simple snapshot of the current situation with no historic temporal or spatial context. To achieve a representative understanding of population variation, this study has shown a requirement for long term data on both the biota of interest and environmental change, and that, ideally, such data is continuous rather than sporadic to account for inter-annual variation and step change in magnitude.

Hydroacoustic data presented in the current study show both a cyclical pattern and inter-annual step change in fish abundance, as measured by single target density. The cyclical pattern described reveals a 6–7 year periodicity between maximum and minimum estimates. A closer analysis shows that apex points vary both in magnitude and their relationship with previous and subsequent years. Step changes, adjacent years with wide disparity in abundance estimate, were found on five occasions. Each occasion had an increase or decrease at least double the adjacent year. The greatest step change was seen between 2007 and 2008 with the latter having an abundance over five times the preceding year. Where sufficient data exist, periodicity is commonplace in biological studies, yet monitoring programmes designed to establish a baseline often fail to meet this basic data resolution requirement. This study demonstrates that the interval during which data are collected will have a significant effect on the understanding of the current state of the population or environment in a historical context. Interpretation that a population is increasing, decreasing or stable, are all possible scenarios for monitoring programmes of less duration than the established periodicity. Management decisions that directly influence significant financial expenditure are too often based on this limited, and likely erroneous, information.

Hydroacoustic monitoring studies, both short and long term, for both lakes [51] and rivers often show an absence of a standardised methodology across all surveys either from different uncalibrated equipment deployed or variation in survey design between years and location. In contrast, the current study is a rare example of the long-term deployment of horizontal hydroacoustics using a standard approach to data collection, for all surveys. Where equipment changes occurred, intercalibration was conducted to preserve data continuity. Quality assurance of data is an essential component to ensure the reliability and quality of survey results particularly where repeatability is required [52]. A standardised approach to hydroacoustic data collection is a desirable component towards confidence in data comparison derived from a regular monitoring programme [4,53].

Other long term regular monitoring of large river multi-species populations in the UK is normally conducted using either electric fishing, netting or angler-catch methods: for instance, the Suffolk Stour [54], which has one of the most comprehensive fish monitoring programmes using electric fishing, with good numbers of sites monitored at regular, frequent intervals over a 40-year period. Data from this programme have also identified clear cycles in abundance of various fish species as well as longer term trends in abundance for others.

*4.2. Fish Abundance and Size Structure of the Population*

Locating fish to establish their spatial abundance is dependent on two factors: the presence of individuals and the efficacy of detection by the chosen method, whether invasive such as electric fishing or non-invasive with hydroacoustics. This study confirms that mobile hydroacoustics is a cost-effective method for river reach level data collection although in common with all sampling methods, an understanding of limitations and challenges is essential to ensure robust, comparable data acquisition. A wide variety of behavioural patterns, with the potential to influence observed acoustic measurements of fish abundance, occur. Within lowland river systems, longitudinal, lateral and vertical movements are common for many of the species considered, on both seasonal and diel scales, and are affected by a wide variety of factors [49]. The life strategy of most fish has a requirement to change location which can be considered a behavioral response to internal and external stimuli acting on an individual. These have the potential to influence observed acoustic measurements of fish abundance.

Temperature is the major factor limiting distribution and behaviour [55] for poikilothermic animals, such as fish. It also plays a critical part in growth, which has direct consequences on the overwintering survival chances of fish; particularly, young of year fish in rivers [56]. Often the most numerous cohort in the fish population in the subsequent year, these fish, where detected, contribute a significant proportion of the total estimated fish density. In this study, it is of particular importance for the hydroacoustic data as the dominant pelagic species, bleak and roach, attain an acoustically 'visible' size in their second year. Flow also has an important role in fish ecology, both direct and indirect. Flowing water brings food, but it also imposes an energetic cost principally from hydrodynamic impacts. Such impacts are ameliorated by the presence of channel habitat that interrupts laminar flow. In regulated lowland rivers such as the Thames, features are often man-made and of limited availability. For many reaches, including those within the study area, the situation is further complicated, particularly where meanders or weir impoundment occurs, as here, the flow dynamics can be minimal and the habitat more akin to a lentic environment. Light is used by fish as a stimulus for timing diurnal and seasonal rhythms. Hydroacoustic surveys on the Rivers Trent and Thames [21,22] observed a significant difference in the spatial distribution of fish between day and night. This variation has also been established in lake environments [57,58]. Applying this evidence, our study followed the best practice approach, conducting all surveys between one hour after dusk and one hour before dawn.

From the available hydroacoustic data, the fish population in this study shows a pattern of spatial heterogeneity often found in other large lowland rivers [59]. Whilst concentrations of fish were present and often associated with in channel habitat features, this association was not universal. Only 40% of the locations with historic high fish density (HDCs) were closely (<100 m) associated with an identified habitat feature (CHIF). The association between high fish density and habitat features was weakest in Reach 4. Here, only 20% of the channel interrupting habitat features were associated with high fish density clusters. However, this reach had the lowest fish abundance in the study area, and therefore the weak association noted may simply reflect the lack of fish present to occupy the available habitat. The hydroacoustic surveys used in this study were all conducted in July. At this time of the year water temperature is relatively high, flows are at or near their lowest, and water turbidity is typically 6–10 NTU, providing a relatively clear water column for a large lowland river. The requirement for active fish in a relatively benign environment to seek shelter is lower than other times of year when reduced physiological activity and strong currents will necessitate the need to seek shelter. As well as longitudinal spacing along a river, the vertical distribution of fish within the water column is critical to establishing a representative knowledge of the population. For hydroacoustics, it is important that surveys are conducted where the target species are 'visible' and not close to riverbed and bank boundaries. Both cyprinids and percids, particularly during juvenile life stages, are associated with mid-water pelagic existence,

especially during the hours of darkness. It is therefore a reasonable assumption that the two most abundant species considered in this study, bleak and roach, are the main contributors to estimating acoustic abundance. Movement away from riverbed and bank boundaries may equally be reflected in distancing from identified habitat features. Environmental conditions during the surveys, combined with the behaviour of those cohorts that contribute a greater proportion of the total estimated abundance, is likely to cause an underestimate of the association between fish and habitat.

Fish aspect is an important factor in determining the proportion of the population acoustically 'visible' within the study reach. Previous studies on rivers show that most fish swim along the longitudinal axis of the river upstream or downstream [60,61]. Using a regression model [40] for converting acoustic size to fish (combined cyprinid and perch) length, a −50 dB minimum threshold setting will detect fish insonified in side-aspect orientation to the acoustic axis at 40.7 mm. Using the approach of [62], application of the von Bertalanffy growth model estimates of fish length at the end of their first year were calculated for the most common species found in the study reach. Scale age data for the Thames indicate that resident fish exhibit relatively slow growth rates, relative to a reference dataset collected from 130 UK river fisheries. From this information, the 'slow' growth curve parameters presented in [62], were selected which indicate bleak at the end of their first year are expected to attain a fork length of 32.1 mm. For roach, chub and perch attainment lengths are 32.0 mm, 39.4 mm and 48.7 mm, respectively. It is therefore assumed that the young of year for these species do not contribute in any significance to acoustic abundance estimates, particularly as the hydroacoustic surveys were normally carried out in July, only part way through the growing season. Capture data from this study show that only 1.32% of bleak and 0.75% of roach were shorter than the acoustic minimum threshold value. However, boom boat electric fishing in a large river environment does not sample very small fish efficiently and will greatly underestimate the absolute and relative abundance of young of year cyprinids. Capture by electric fishing of 0+ fish will be biased towards the larger members of the cohort and so it is not possible to estimate the true size-structure of that cohort

A recent study [63] describes the effect of target strength oscillations, generated by surface or bottom-induced sound multi-pathing, as a potential source for serious errors in estimates of fish abundance and biomass in horizontal acoustic surveys of extremely shallow inland waters (depth 1.7–2 m). Errors can be reduced by avoiding phase boundaries, restricting the maximum usable range (MUR) to ~10 m from the transducer face, and using narrow beams with low side-lobes. The dual-beam deployed from 1994–2002 had circular beam opening angles of 6°/15° (narrow/wide beams), and analysis ranges were up to 24 m. The split-beam equipment used from 2002 included low side-lobe 4° × 10° elliptical transducers and is better designed for use in shallow waters. Average MUR ranged from 8.8 to 10.9 m (from 2015 survey) over the four reaches and the section of the River Thames covered in this study is predominately greater than 2.5 m deep, thereby reducing the likelihood of TS oscillations and gross errors in fish abundance estimates.

*4.3. Divergence of Results from Marginal Electric Fishing and Mid-River Hydroacoustics*

Within the study period, patterns of overall fish abundance derived from boom boat electric fishing surveys of margin waters, differed from those from the mid-river hydroacoustic surveys, contrasting with other periods, where broad agreement occurred.

We consider that the variation in spatial distribution between margin and mid-river of roach and bleak as shown from electric fishing captures is a plausible reason for the difference we see in patterns of overall fish abundance derived from the hydroacoustics (that primarily samples mid-river, mid-water) and from the margin boom boat surveys. In most years, the mid-river fish community is dominated by bleak, and therefore, a likelihood that estimates of overall fish abundance from hydroacoustics will diverge from those estimated from margin electric fishing. Boom boat operators have observed large

numbers of bleak, avoiding the channel-side limit of the electric field and veering into the middle of the river [64]. This would suggest that overall bleak abundance in the Thames is significantly underestimated in the margin electric fishing CPUE results. In contrast, bleak were the dominant species mid-river in precisely the location targeted by acoustic survey, further compounding the differences in the two abundance estimates.

Length-frequency data from margin fished boom boat data, weighted according to total fishing time, indicate annual patterns of abundance for bleak and roach are very similar, notwithstanding that in most years, roach catches are higher than those for bleak. This disparity in capture numbers is expected, the electric fishing data clearly showing a higher proportion of roach captures in the river margins. The years 1994, 1995 and 2016 were good recruitment years for both species. Difference in cohort strength were also noted with roach showing strong recruitment in 2004 and 2005. For bleak, the identification of prominent year classes based on length-frequency from electric fishing catches is a challenge in the Thames as the species is prolific in most years, unlike some species that only recruit strongly every so many years. Whilst we were able to determine from the data that 2013 and 2017 were good recruitment years, cohort strength is less variable than for longer lived species, including roach. For bleak, we contend that influences on survey data from sampling environmental conditions such as wind speed, rainfall, moon phase, illumination, are at least as important in acoustic abundance estimation. Boat avoidance factors and minimal variation in acoustic beam orientation are also sources of influence on data collection. Thirdly, instantaneous spatial distribution both vertically in the water column and longitudinally along the river of a highly mobile and shoaling species are also important considerations. Caution with data interpretation is required due to variation in spatial preference between the two species. Whilst roach dominate margin locations from where recruitment assessment data is derived, bleak show a distinct spatial bias towards mid-river. The impact of sub-optimal sampling habitat for bleak is considered a potential source of error particularly for representative cohort apportionment.

Scale age data from the study indicate that for the early cohorts, roach and bleak are similar in length. Electric fishing catches are dominated by fish of between 70 and 120 mm length, with growth analysis indicating bleak ages to be 2+, 3+ and 4+. Roach catches are largely dominated by 3+ and 4+ fish. A comparison of the length-frequency growth information with scale age data suggests that these species are not sampled representatively by boom boat electric fishing until three or four years old. In contrast, the hydroacoustic surveys will detect small cyprinids c. 40 mm, i.e., 1+ fish. The 1+ roach and bleak will therefore present in the hydroacoustic surveys perhaps two or three years before the same cohort is caught efficiently by the boom boat. Within a balanced fishery, these early cohorts will represent a high proportion of the total fish population and their presence or absence likely to have a significant impact on abundance estimates. We hypothesise that the lag in the peaks of abundance noted in this study between the results from the two survey methods is due primarily to size-selectivity rather than species selectivity.

### 4.4. What Might Have Driven the Apparent Increase in Total Fish Abundance in the Late 2000s?

Large rivers across Europe, such as the Thames, have been modified for a range of reasons including flood mitigation and navigation. Hydrological connection, where present, is typically in the form of artificial canals or man-made lakes constructed for navigation, drainage or fisheries. Such interventions commonly result in a channelised river largely separate from its natural floodplain [65]. A study [66] on the impact of human pressures on fish assemblages found that even eurytopic species, such as roach and bleak, considered relatively robust to anthropomorphic environmental pressure, are impacted negatively in that situation. Increased velocity in river margins with impoverished habitat limits recruitment opportunities, and at times of flood, particularly in the colder months,

there is an increased likelihood of downstream drift and washout, especially for juvenile life stages. In rivers such as the Thames where these conditions exist, habitat features that provide structure to mitigate flow velocity extremes are important for fish recruitment.

Occasionally, most often during winter, 'out of bank' flood events result in a temporary reconnection between the main channel and its floodplain. By comparison, summer 'out of bank' conditions are relatively rare events. During the twenty-five years of this study, ten 'out of bank' events were recorded, of which only a single event occurred during the summer months. River channel connection with its floodplain persisted for just ten days during this unique event. Relating flow and temperature events to hydroacoustic fish densities; the summer 2007 inundation was followed by average winter flows and temperatures, with very high fish densities in subsequent years. In contrast, the 2012 event remained in bank and was followed by cold winter floods, with fish densities in subsequent years showing no obvious increase.

In relatively unmodified floodplain rivers, in temperate regions, high spring and summer flows that result in prolonged floodplain inundation are considered beneficial for fish recruitment. Floodplain water bodies provide warm, shallow, low velocity habitats, ideal for spawning of phytophilic and eurytopic species and for early larval and fry growth of all species [67–69], and often rich in food items emanating from nutrients in floodplain soils.

In heavily modified rivers where the floodplains are largely disconnected from the main channel, such summer flood events can, perversely, have negative impacts on fish communities. Where rivers are straightened and embanked, water velocities at high flows will rise often far beyond the swimming capabilities of juvenile fish before the banks overtop and floodplains are inundated. When fish can access the flooded land, they may die due to asphyxia from terrestrial vegetation that dies and rots when submerged for more than a few days; in addition, any surviving fish may not be able to move back into the main channel when waters recede, due to flood banks and associated control structures [70]. However, where floodplain inundation occurs in a way such that fish are able to access the floodplain on a rising river, and where natural, unimpeded drainage back to the main channel is possible as the flood recedes, in such instances, summer floods can be beneficial and fish species may take opportunistic advantage to boost recruitment. It is hypothesized that this may have happened on the middle Thames in 2007 and resulted in a general increase in the abundance of roach and bleak, in particular, in the years following the summer 2007 floods. This is in contrast to other rivers, where fish populations declined significantly in the wake of that event and remained generally lower than previously, such as the Nidd in Yorkshire, and the Upper Thames where there was a decrease in overall fish abundance in mid-late 2000s followed by a recovery [71].

This study shows a clear increase in annual hydroacoustic fish abundance in the three years after the 2007 'out of bank' summer event. This response is most obvious in those reaches (1–3) where floodplain access is greatest. Flood risk modelling [72] indicates that the river between Reach 1 and Reach 3 has substantially greater spatial opportunity for floodplain connection as measured by floodplain hectare per river kilometer during high (1:20 and 1:100) flow events compared to Reach 4 (43–65% of the others). Natural topology in Reach 4 also results in greater constraint of the river channel when compared with the upstream reaches. The difference in floodplain access correlates with the inter-reach variation, in response to the 2007 summer 'out of bank' event. Data for Reach 4 clearly show a suppressed response in subsequent years (2008–2010) when compared with the fish density increase seen in the three upstream reaches. However, age and length data, from electric fishing surveys, do not support the assertion that 2007 was particularly good for recruitment, rather that there were strong year classes originating in the early 2000s. In addition to this summer flood event, the other environmental factor considered is overwintering survival. Both preceding and subsequent winters were of average temperature and flow when compared to the entire study period; conditions therefore would not likely convey any obvious advantage or disadvantage when compared to other

years in the study period. We tentatively hypothesise that the summer 2007 floodplain inundation may have benefited cohorts of roach and bleak from preceding years by providing rich summer feeding areas, indicated by increases observed in acoustic abundance during subsequent years. Determining the role and relative influence of these various environmental factors on cohort recruitment and survival, requires further site-specific study.

### 5. Conclusions

Understanding both the value and limitations of data collected to meet environmental monitoring requirements are critical considerations for an appropriate application to evidence-led fisheries management.

This study demonstrates the practicality and benefits of long-term, standardised hydroacoustic surveying supported by supplementary boom-boat surveys in large, managed, lowland river systems where other methods are unsuitable and destructive sampling is unacceptable. This in turn has identified cyclical patterns in fish abundance with long periodicity previously identified in small UK rivers.

This Thames time-series dataset will provide local fisheries managers with a comprehensive baseline to determine potential impacts of local and national infrastructure projects on fish populations, such as large-scale abstraction for storage (Thames Water Abingdon Reservoir Proposal), water transfer schemes (Severn-Thames Transfer) and low-head hydroelectric facilities such as the recently commissioned Archimedes turbines at Culham. It is also hoped that the benefits of large, landscape-scale floodplain improvements, such as the Earth Trusts two 'River of Life' schemes, can be quantified.

In determining future survey work, the current study offers broad guidance on existing knowledge gaps and where targeted small-scale high-resolution sampling may be beneficial. Further research is recommended on the hypothesised benefits of summer inundation events accrued under conditions of longitudinal connectivity with subsequent benign winter flows and temperatures.

**Author Contributions:** Conceptualization, J.L. and J.H.; methodology, J.L., J.H., G.P., F.E. and S.M.; formal analysis, J.L., J.H., G.P., F.E., S.M., K.T.; writing—original draft preparation, J.L.; writing—review and editing, J.L., J.H., G.P., F.E., S.M., K.T.; visualization, J.L. All authors have read and agreed to the published version of the manuscript.

**Funding:** This research was funded by Environment Agency, UK.

**Institutional Review Board Statement:** Not applicable.

**Informed Consent Statement:** Not applicable.

**Data Availability Statement:** All data are curated and stored in databases of the Environment Agency, UK and can be made available on request to the authors. All requests will be subject to the UK Data Protection Act 2018.

**Acknowledgments:** The authors wish to thank Jonathan Baxter, Andy Killingbeck and numerous other Thames West fisheries staff for their help with providing TWARP data and local field knowledge. Matt Loewenthal (EA), Katherine Dolman (EA), Andrew Davies (EA) and Mike Bowes (CEH) for providing data and modelling on temperature and flow. Assistance with GIS modelling was provided by Mike Shankster (EA). We are grateful to two anonymous referees for their constructive reviews improving the manuscript.

**Conflicts of Interest:** The authors declare no conflict of interest.

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
