# Peer review of "An Assessment of Hydroacoustic and Electric Fishing Data to Evaluate Long Term Spatial and Temporal Fish Population Change in the River Thames, UK"

_water, doi:10.3390/w13202932_

Round 1

Reviewer 1 Report

Thank you for the opportunity to review the paper “An assessment of hydroacoustic and electric fishing data to evaluate long term spatial and temporal fish population change in the River Thames, UK”.

Overall, the paper is very complex, very well structured and well written.

However, there are some concerns that need to be addressed.

First, in introduction, I suggest the authors to highlight the originality/ uniqueness of the paper. In addition, I suggest the authors to add more references related to this topic.

Second, I suggest the authors to add (after Conclusions) the main limitations of the study and some future studies recommendations.

Good luck!

Author Response

Dear Reviewer,

Thank you for your constructive reviews on improving the manuscript. We have addressed your comments as follows:

  • A graphical abstract is provided (including a JPEG version).
  • Introduction now highlights the originality/uniqueness of the study. This is supported with additional, recent references.
  • Full contact details are provided for all equipment manufacturers and software suppliers where first cited in the paper.
  • The discussion section has been re-worked to consider and discuss more fully the study results. Additional, recent references are provided. General text has been edited, its use to frame the specific findings and provide a wider context.
  • Concluding remarks are both specific to the study findings and also describe the limitations of the study. Future work suggestions are provided.
  • Reformat of all table titles to locate above the relevant table.
  • The funding source is provided.
  • Details (name, contact details, paper contributions) for the sixth author are provided.

Regards,

Jim.

Reviewer 2 Report

Manuscript ID water-1393575 entitled "An assessment of hydroacoustic and electric fishing data to evaluate long term spatial and temporal fish population change in the River Thames, UK" in my opinion is an interesting and valuable research material. Researches concerned the results from mobile hydroacoustic surveys and electric fishing boom boat carried out between 1994 and 2018, to assess the fish stocks in four impounded reaches, covering 19.8 km of the River Thames. I believe that this is an important issue from the point of view of ecology, environment science and fish economy.

It fully corresponds to the profile of Water journal. It is written in the correct scientific and technical language, its organization and division into chapters is correct, the analysis of the obtained research results has been properly carried out and correct summary have been formulated. In my opinion, it may be published, and the comments below are left to the Authors:

  1. A graphical abstract would be very useful for the readers and increase the attractiveness of the publication.
  2. I believe that the introduction and discussion should refer to the current literature, as most of the cited items are quite old.
  3. The manufacturers of the equipment used should be listed. This needs to be completed.
  4. Acronyms and abbreviations chapter may be introduced at the end of the manuscript.
  5. Founding source should be given.
  6. The conclusions are very general and do not relate to the results obtained during the conducted research. This should be corrected.

Author Response

(The authors gave the same response as above.)
